# Relationship Between Effective Dose, Alternative Metrics, and SSDE: Experiences with Two CT Dose-Monitoring Systems

**DOI:** 10.3390/diagnostics15131654

**Published:** 2025-06-28

**Authors:** Lilla Szatmáriné Egeresi, László Urbán, Zsolt Dankó, Ervin Balázs, Ervin Berényi, Mária Marosi, János Kiss, Péter Bágyi, Zita Képes, Miklós Emri, László Balkay

**Affiliations:** 1Division of Radiology and Imaging Science, Department of Medical Imaging, Faculty of Medicine, University of Debrecen, Nagyerdei St. 98, H-4032 Debrecen, Hungary; urban.laszlo@med.unideb.hu (L.U.); danko.zsolt@med.unideb.hu (Z.D.); balazs.ervin@med.unideb.hu (E.B.); eberenyi@med.unideb.hu (E.B.); marosi.maria@med.unideb.hu (M.M.); kiss.janos31@med.unideb.hu (J.K.); 2Doctoral School of Molecular Medicine, Nagyerdei St. 98, H-4032 Debrecen, Hungary; balkay.laszlo@med.unideb.hu; 3Doctoral School of Neuroscience, Nagyerdei St. 98, H-4032 Debrecen, Hungary; 4Iconomix Kft. Koller St. 9B 5., H-7626 Pécs, Hungary; 5Division of Nuclear Medicine and Translational Imaging, Department of Medical Imaging, Faculty of Medicine, University of Debrecen, Nagyerdei St. 98, H-4032 Debrecen, Hungary; kepes.zita@med.unideb.hu (Z.K.); emri.miklos@med.unideb.hu (M.E.)

**Keywords:** computed tomography (CT), dose-monitoring system (DMS), effective dose (ED), size-specific dose estimate (SSDE), size-specific effective dose (SED)

## Abstract

**Background**: We assessed the frequency and causes of discrepancies in CT dose indices such as dose-length product (DLP), size-specific dose estimate (SSDE), and effective dose (ED), as calculated by CT dose-monitoring systems. Our secondary aim was to demonstrate the estimation of size-specific ED (SED) from the patients’ dose records. **Methods**: The retrospective study included dosimetric data of 79,383 consecutive CT exams performed on two CT scanners. The following dose values were recorded from both the locally developed dose-monitoring system (DMS) and a commercial dose-monitoring program (DW^TM^): DLP, SSDE, and ED. Only the DMS provided bodyweight-corrected effective dose (SED_DMS_) and the SED based on previous published data. **Results**: Without body-region-specific analysis, there were no tendentious differences between the DLP, ED, or SSDE values obtained from DW^TM^ and DMS. However, the body region-based correlation revealed substantial differences between ED_DMS_ and ED_DW_, primarily related to inadequate identification of the body. SSDE showed strong correlation to each anatomical category and CT device, except for the head region, where inadequate consideration of CT inclination was the reason for the biased SSDE_DW_ value. Furthermore, by analyzing the SED_DMS_, SSDE, and SED correlations, we concluded that SED_DMS_ is a promising figure for estimating the SED value. **Conclusions**: SED provides suitable supplementary size-specific dose data to SDDE and may be a preferable choice for estimating cumulative doses in routine radiological practice.

## 1. Introduction

In accordance with the EURATOM European Directive [1], all departments in the European Union using medical ionizing radiation must ensure the dose optimization of radiological procedures and provide up-to-date documentation on patient radiation exposure. Several types of commercially available dose estimation software, such as Radimetrics^TM^, Dosewatch^TM^ (DW^TM^), DoseWise^TM^, Teamplay^TM^, DoseTarck^TM^, Qaelum^TM^, and DoseMonitor^TM^ [2,3,4,5,6], open source packages (e.g., OpenREM), and locally developed dose management systems can help in documenting and monitoring radiation doses in medical imaging. We have used both a locally developed dose-monitoring system (DMS) and a commercial application (DW^TM^) in our department since 2020.

Direct dosimetric indices such as CT dose index (CTDI) and the dose-length product (DLP) are routinely stored in the “dose information page” or “dose report” CT files [4]. In addition to these conventional metrics, the effective dose (ED), a dose quantity related to the risk of stochastic effects, is another commonly applied measure and can be calculated using the following formula [7]:ED (mSv) = DLP (mGy × cm) × f (mSv/mGy × cm),(1)
where f is the conversion factor from DLP to ED. The age and body region dependent f values are determined using the published tissue weighting factors for the ICRP (International Commission on Radiological Protection) reference phantom [8,9]. Since this constant only applies to the average patient size, ED cannot be estimated for different patient sizes and/or body weights with this formula.

Another aspect is that for the same amount of radiation (CTDIvol, DLP), the organ doses decrease as the patient size increases. A 2011 study employed a specific mathematical model to describe how relative doses vary with patient weight [10]. To account for this variation, the concept of patient-specific or size-specific ED, referred to as SED, was introduced recently [11,12]. The SED, a derivative of ED, is calculated using a digital phantom adjusted to the patient size and takes the total organ doses and the associated tissue [9] weighting factors into account. The aforementioned digital phantom technique is, in fact, a validated and complex Monte Carlo-based CT simulator, which is an increasingly common solution [11,13,14,15]. SED can be also obtained by incorporating a weight-adjusted factor (wkg) into the ED formula for patients outside the reference size [16,17]:SED = wkg × f × DLP.(2)

The CTDIvol, DLP, and ED are estimated for the reference 16 cm (head) and 32 cm (body) phantoms, which do not conform with most patient sizes and body morphology; in contrast, the SED considers the patient size with the wkg factor. The currently accepted size-related dose metric is the “size-specific dose estimate” (SSDE), which was proposed by the American Association of Physicists in Medicine to convert volume CTDI into patient-size normalized doses based on patient cross-sectional diameter (such as anterior-posterior, lateral, or effective diameters) [18,19,20]. Although either patient effective diameter or the water-equivalent diameter can be used to estimate SSDE, the latter considers the actual radiation attenuation properties of the body tissues, and thus, it provides a more accurate and realistic result [19]. The linear relationship between SSDE and SED [12,21,22] suggests that SED can be estimated from SSDE. In principle, this could support the development of a sufficiently accurate lookup table to derive SED values from SSDE.

Based on the above considerations, the present study set out to achieve three objectives. First, compare the CT dose estimates, including DLP, ED, and SSDE, from the DW^TM^ and the local DMS tools. Second, investigate the differences between ED, weight-corrected effective dose (SED_DMS_), and SSDE for various body regions and CT scanners. Third, compare SED_DMS_ against SED generated from the Monte Carlo simulation reported by Martin et al. [11].

## 2. Materials and Methods

The Regional Ethics Committee authorized the protocol of our retrospective study. The study was performed in the Department of Medical Imaging, Faculty of Medicine, University of Debrecen (Debrecen, Hungary) (Permission number: DE RKEB/IKEB 6732-2024). We connected the local DMS and DW^TM^ software (GE Healthcare, Chicago, IL, USA; version 3.1.5) with two CT scanners (GE Healthcare, Chicago, IL, USA) in our department (GE Revolution HD [CT1] and GE Revolution Evo [CT2]). Our study included DMS and DW^TM^ dose data for all CT exams performed between June 2020 and December 2023 on these scanners.

From the data collected, we excluded patients under 15 years of age, CT examinations performed within the framework of clinical trials, extremity CTs, and CTs with poorly written protocol names. We also excluded CT exams without a specified “target area” (defined as “undefined region”) and whole-body CT examinations. Eventually 79,383 CT exams of patients aged between 16 and 104 years (mean age: 63.3 ± 15.3 years) were included in our retrospective analysis (female: 41,188; male: 38,117; sex not defined: 78). In this work we use “CT scan” as a synonym of “CT study”, and of course a CT scan can consist of multiple series.

The DW^TM^ supports the data export into a Microsoft Excel™ file. The data from the MS SQL server-based DMS were exported in Microsoft Access. We filtered and merged the corresponding DW^TM^ and DMS CT dose data into a single MS Access^TM^ database and performed all data evaluations and processing in MATLAB^TM^ (The MathWork Inc., Natick, MA, USA; version R2024b). DW^TM^ used the conversion factors published by Deak et al. [8] to estimate ED; the DMS used the conversion factors recommended by the American Association of Physicists in Medicine (AAPM) [20].

The ED values of the DW^TM^ (ED_DW_) and DMS-derived ED (ED_DMS_) and SED (SED_DMS_) were determined using the following equations:ED_DW_ = f(Deak 2010) × DLP(3)SED_DMS_ = wkg × f(RPT 96) × DLP(4)ED_DMS_ = f(RPT 96) × DLP(5)

For SED_DMS_, we applied a weight-adjusted w_kg_ factor for estimating weight-corrected ED from DMS using data from the prior publication of Huda et al. [10]. We used the following formula for wkg calculation [16], where w represents patient weight in kilograms:wkg = 1.73 − 1.33 ∙ 10^−2^w + 4.04 ∙ 10^−5^w^2^(6)

The DMS estimated SSDE from patients’ effective diameter (Deff), CTDIvol, and the formula suggested by the AAPM [20]. We also calculated SSDE (Dwater) from the water-equivalent diameter (Dwater) of thoracic regions based on the published relationship between SSDE(Dwater) and SSDE(Deff) on chest CTs [23].

Due to the large number of protocols (>100) on the two CT scanners, interpretation and analysis of the protocol-based comparison of DW^TM^ and DMS was complicated and challenging. Hence, we classified all protocols into six anatomical regions (as “alias”): head, neck, chest, abdomen, pelvis, and trunk.

The ED and SSDE values were available in different ways in the two dose-monitoring programs. The ED_DW_ values were accessible only at the study level (the total ED was stored in each series), while the SSDE values were available for each series. In contrast, ED_DMS_ and the SSDE_DMS_ values were available at both the study and series levels. To account for this discrepancy, we introduced two different types of regional anatomical terminologies, study-based and series-based aliases. The aliases, in the case of studies (total of 79,383), included head (22,051), neck (1378), chest (40,810), abdomen (14,541), and pelvis (603) names. The series-based region names defined in the DMS for each series including scouts and bolus tracking images were defined by the CT radiographers and had 93,259 series, including head (18,818), neck (5171), chest (45,041), abdomen and pelvis (29,259), and the trunk region (the chest, the abdomen, and the pelvis: 7652). The series stored in DW^TM^ and DMS were matched based on the DICOM series instance ID.

Finally, we estimated the SED from the CT scans based on Martin et al. [11,13], who calculated SED on a digital phantom using Monte Carlo simulation and adjusted it to the patient size, taking the total organ doses and associated tissue (ICRP 2007) weighting factors into account. We calculated SED values from the SED-SSDE correlation equations ($SEDs=\left(a\times SSDE\right)-b$) for specific regions and scan lengths, as described in a prior publication [11]. Our selection of body regions and scan lengths to match the prior publication [11] led to the creation of the following regions: chest, abdomen-pelvis, chest-abdomen-pelvis (CAP), abdomen, and chest-abdomen. The scan lengths and the number of cases are summarized in Table 1.

## 3. Results

Figure 1 presents box plots of ED, SED_DMS_, and DLP values for the whole study population calculated by DW^TM^ and DMS.

Figure 2a,b presents the results of the region-based correlation analysis of ED data for the 79,383 studies for the two CT scanners separately for DW^TM^ and DMS. The region-based grouping of the ED values resulted in one cluster for each anatomical area (indicated with different colors). Compared to the CT2 scanner, the correlations were stronger for CT1 and are closer to the desirable slope of 1. For CT2, there are several distinct populations for a given region, and the correlation, for example, in the chest or the neck CT exams, shows a significant scatter. Although the association is strong in the head region, the slope deviates markedly from the 45-degree straight line.

Although we did not specifically assess the cause of variations in correlations, the inclusion of multi-series CT protocols for different body regions (particularly for abdomen-pelvis CT) might explain the variable correlations. Since DW^TM^ assigns the same region for each series within a study, it likely calculates incorrect ED for multi-series CT exams. To bypass this issue, we excluded CT exams with more than one body region (Figure 2c,d).

Figure 3 illustrates the differences in assigning the correct body regions with the two dose-monitoring software platforms for the same CT study.

The DMS automatically populates the target region and can be edited manually, versus the fixed, non-editable DW^TM^ target regions, which are identical across all series. The possible appearance of the fixed target area for all series in the DW^TM^ system is illustrated in Figure 3c. In the sample study, both head and chest examinations were performed, consisting of two localization scans (scouts), an unenhanced head and an unenhanced chest CT. The DW^TM^ assigned all scans to the chest region, resulting in an effective dose of 19.24 mSv. With the DMS, each body region is classified accurately (head and chest), resulting in a substantially lower total effective dose of 3.96 mSv (Figure 3a). The manual reassignment of the body region in the DMS per the target regions of DW^TM^ (chest-chest, as in Figure 3c) resulted in a total effective dose of 19.96 mSv. The small difference between the total doses (19.24 vs. 19.96.) is explained by the distinct f factors used in DW^TM^ [8] and DMS [20].

Figure 4 highlights strong but variable correlations between SSDE_DW_ and SSDE_DMS_ for different CT series.

Figure 4 reveals a strong correlation between the SSDE data of DW^TM^ and DMS for almost all anatomical regions. To better explore SSDE correlations, region-based SSDE analyses were also performed for both CT1 and CT2 (Figure 5). The strongest correlation between SSDE_DW_ and SSDE_DMS_ was found for neck (y = 0.988x − 2.25, R^2^ = 0.987), chest (y = 1.014x − 0.00685, R^2^ = 0.987), abdomen-pelvis (y = 1.00x + 0.116, R^2^ = 0.955), and trunk (y = 0.9827 + 0.2776, R^2^ = 0.9572). In contrast, the head region (y = 1.166x-2.609, R^2^ = 0.7908) had the weakest correlation, with higher SSDE_DW_ compared to the SSDE_DMS_ data.

Apart from the above correlations, we examined the associations between SSDE and ED. The relationships ED_DMS_-SSDE_DMS_ and SED_DMS_-SSDE_DMS_ for different body parts and CT scanners are depicted in Figure 6 and Figure 7.

The SSDE values of chest and chest-abdomen CT exams were recalculated using water equivalent diameter based on the results of Michele and colleagues [23]. For all other anatomical regions, SSDE values were based on the effective diameter as initially calculated by DMS. Table 2 summarizes the linear fitting results for each body region and CT scanner. Figure 8 presents correlations between DMS-calculated mass-normalized ED (SED_DMS_) and SED.

## 4. Discussion

Our study demonstrates excellent correlation between DW^TM^- and DMS-derived DLP and the SSDE, implying the suitability of the in-house system for CT radiation dose monitoring (Figure 1 and Appendix A). Although a trend difference between the median (or mean) of mass-corrected and uncorrected ED statistics was expected, such a distinction was not observed. As noted in Appendix A, the range and the standard deviation of the SED_DMS_ values were substantially smaller than those of the ED_DMS_ values, implying that the distributions were not identical.

The relationship between ED_DW_ and ED_DMS_ (Figure 2a,b) revealed several highly linearly correlated populations, despite substantial variability based on the scanner type (CT1 vs. CT2), body region, and study complexity. Additionally, the straight lines do not align exactly with the 45-degree reference, which could be attributed to the different f factors used for the calculations. The DMS used the factor f recommended by the AAPM [20], whereas for DW^TM^-based calculations we used the factor from Deak et al. [8]. The presence of populations falling far from the 45-degree lines could be related to the body region identification method of DW^TM^ for the CT series. The DW^TM^ was particularly limited for studies with different body regions, especially when the CT series included incorrect and unmodifiable body regions, leading to erroneous ED calculations. Such errors stemmed from two body regions (such as head and chest CTs) imaged during single CT examinations, where both scanners (CT1 and CT2) assigned the body region corresponding to the last series to all previous series. Improvements in correlation were observed following the exclusion of studies with a modified protocol compared to the basic protocol (e.g., emergency CT cases) (Figure 2c,d). Nevertheless, the plot for CT2 still included two to three populations (the chest and head) with slopes that were not close to one. In these populations, there were several specific CT2 protocols with different regions, where the DW^TM^-derived ED values were biased. Some representative cases are presented in Figure 3 and in the Appendix A.

Figure 3 and Figure 4 generally show an excellent correlation in the SSDE comparison, except for the skull region. We found that the discrepancy between the two programs could be related to the incorrect determination of the central slice by the DW^TM^ program. Per Juszczyk et al. [24], DW^TM^ malfunction for detecting the central slice could be related to the tilted gantry at the central slice at a different location compared to the actual central slice of the DW^TM^ (Figure 9).

The SSDE is currently the preferred dose metric for CT because it inherently includes patient size information. Despite its widespread adoption, the application of SSDE is subject to some limitations. For the calculation, an on-site software application (proprietary software or dose manager) is required, as most CT manufacturers do not usually provide the SSDE data of the series in the DICOM dose report file. To overcome the challenges of geometric size calculation, some researchers have introduced body mass index (BMI) as a size metric to replace mid-slice effective diameter in SSDE determination [25,26,27]. In addition, the SSDE method has also been adopted for DRLs (diagnostic reference level); however, the age and size ranges of SSDE-related DRLs are inconsistent across publications [28. The inclusion of SSDE in DRLs remains challenging, since DRLs depend not only on age and body region but also on the actual scan length.

Our evaluation on the relationship between SSDE_DMS_ and ED_DMS_ (Figure 6 and Figure 7) supports prior evidence suggesting that SSDE provides good input data for estimating SED, as it includes patient site data [11,12,21,22]. There was an acceptable linear correlation between SSDE_DMS_ and ED_DMS_, while SSDE_DMS_ and SED_DMS_ demonstrated a strong association in each body region. SED_DMS_ values were also derived from the SSDE_DMS_ data based on the slope and the intercept of the different body parts as determined by Martin and co-workers [11]. The SED data correlate very well with the SED_DMS_ values (Figure 8). Moreover, the slope is very close to one for most body regions. Table 2 presents the resulting fitted linear equations, along with the corresponding R^2^ values and the percentage differences in slopes, using the slope from [11] as the reference. We observed variable correlation (0.452–0.949) and percentage differences between ED and SSDE (Table 2), suggesting the discrepancies are not fundamentally related to the variance of data points. The mean (and the range) absolute percentage difference of ED_DMS_-SSDE_DMS_ was always higher at about 28% (12–65%) than that of SED_DMS_–SSDE_DMS_ [11.6% (4–29%)]. Thus, the theoretical SED can be better estimated using the weight-corrected ED (SED_DMS_) rather than the ED (ED_DMS_). There are at least two reasons why distinct CT scanners have different slopes and intercept parameters. First, the scan lengths in our data across body regions and CT scanners were variable and different from the presumed scan lengths used in the previously reported theoretical SEDs [11]. Second, our DMS does not provide D_water_-based SSDE and is therefore inherently biased compared to the SED calculated by Martin and coworkers.

Our results are concordant with previous publications [12,16] in reinforcing the value of SSDE to reliably approximate patient-specific radiation risk when estimated from effective or water-equivalent diameters. This finding is particularly relevant for most centers that lack access to Monte Carlo-based dosimetry tools. Further research is, however, needed to refine the wkg factor and find a more reliable body mass or BMI function. Furthermore, a key advantage of ED or SED applications is their utility in comparing radiation exposures from various sources [28,29].

The present study offers valuable data to the CT patient dose research community. Firstly, processing and analyzing more than three and a half years of CT examinations led to the assessment of 79,383 CT images in total, constituting a large sample size. This enabled a comparative evaluation of two software programs that contributed to the improvement and validation of our own DMS software, providing more profound insight into the daily dose management in our department. Secondly, the study-based comparison of the ED from the two programs and various correlation analyses of SSDE data made it possible to uncover unknown aspects of the locally established DMS, which could serve as a basis for its long-term development. Thirdly, to achieve adequate data interpretation and the seamless functioning of radiation dose monitoring, it is imperative to establish a dose optimization expert team of medical physicists, radiologists, radiographers, and IT specialists [30,31,32].

Fourthly, our findings indicate that SED_DMS_ has several advantages over ED, and therefore, the weight-corrected ED parameter may be even more suitable for estimating the cumulative dose for patients. The burgeoning use of CT scanning has brought renewed attention to the issue of cumulative doses, necessitating the introduction of a metric that allows for the summation of individual doses [33,34,35]. Although ED remains the most widely used metric for this purpose, a new dose reference level parameter has been proposed (defined as recurrent exposure reference levels) [36,37]. Given the inherent advantages of SED over ED, SED will likely allow better risk estimation.

Despite the above strengths, our study has several limitations. First of all, our work did not extend to examine organ doses. Another limitation is that contrary to recent practices that link dose data to clinical indications (such as pulmonary embolism or urolithiasis), we grouped the clinical indications into different body regions [38,39,40,41]. Although we plan to include the clinical diagnosis data in our in-house CT dose database, the elaboration of this option remains part of future work. A further limitation is that in the current DMS, the SSDE is calculated based on Deff rather than the Dwater parameter, as the DMS was not designed to store images of the CT series (which would be needed to calculate Dwater). In the next step, we will improve DMS to allow for the determination of Dwater.

## 5. Conclusions

In summary, our in-house dose-monitoring system allows estimation of DLP, SSDE, and ED comparable to that of the commercially available DW^TM^ software. Regarding the determination of ED in DW^TM^, errors occur due to the incorrect identification of body regions. In the case of skull CTs, calculating SSDE is problematic if the algorithm does not correctly handle the images from a tilted CT gantry. We propose that SED_DMS_ may be a potential candidate for calculating cumulative doses and estimating the risk-related size-specific effective dose.

## Figures and Tables

**Figure 1 diagnostics-15-01654-f001:**
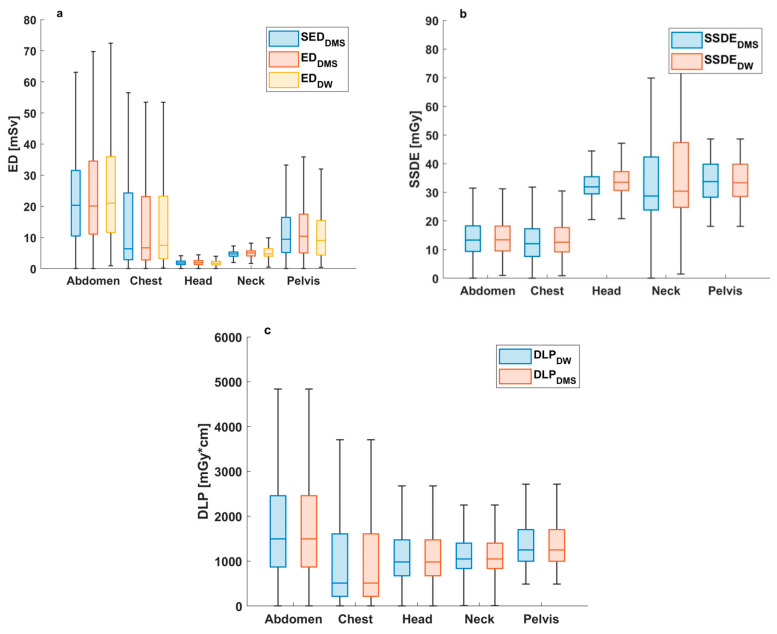
Box and Whiskers plots summarizing ED (**a**), SED_DMS_ (**a**), SSDE (**b**), and DLP (**c**) statistics. A total of 79,383 exams were analyzed for ED and DLP, while 93 259 series were included for SSDE. For ED, DLP, and SSDE values, subscripts indicate the software from which the data originated. The bottom and top edges of each box represent the 25th and 75th percentiles, respectively, and the middle mark represents the median. The whiskers extend to a distance of 1.5*IQR from the top or bottom of the boxplot, where IQR represents the interquartile range. (Key: DW^TM^: DoseWatch^TM^; DMS: locally developed dose-monitoring system; ED: effective dose; SSDE: size-specific dose estimate; DLP: dose-length product).

**Figure 2 diagnostics-15-01654-f002:**
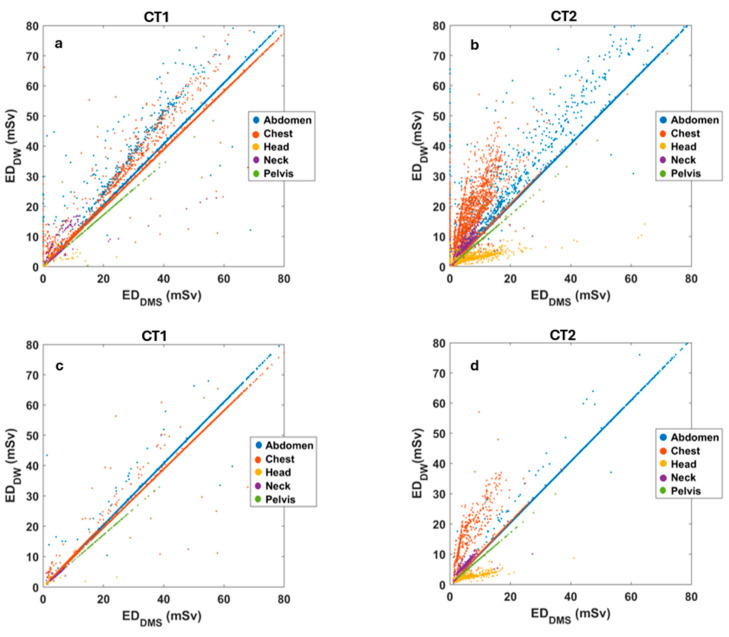
Scatter plots of ED values estimated by DMS and DW^TM^ servers for CT1 (**a**,**c**) and CT2 (**b**,**d**), respectively. The body regions are marked with dots of different colors.

**Figure 3 diagnostics-15-01654-f003:**
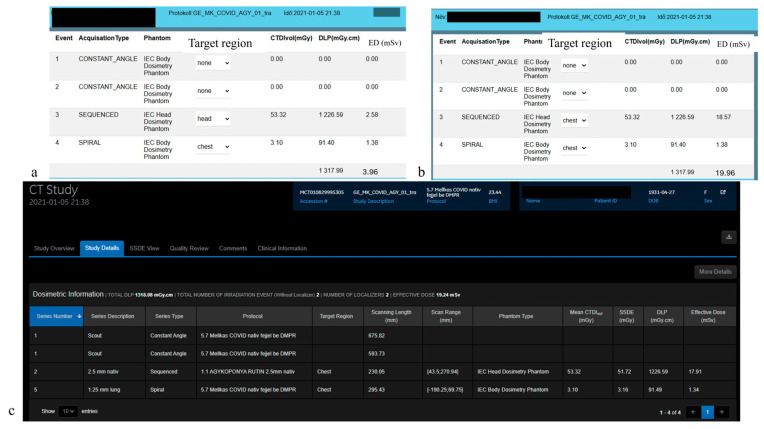
Web interfaces of dose-monitoring systems. Panels (**a**,**b**) present the DMS dose report of a representative study, while panel (**c**) displays the DW^TM^ report of the same study. The (**a**,**b**) panels show the correctly and incorrectly selected region names for the same CT study. All three panels display two localization scans, an unenhanced head series and an unenhanced chest series.

**Figure 4 diagnostics-15-01654-f004:**
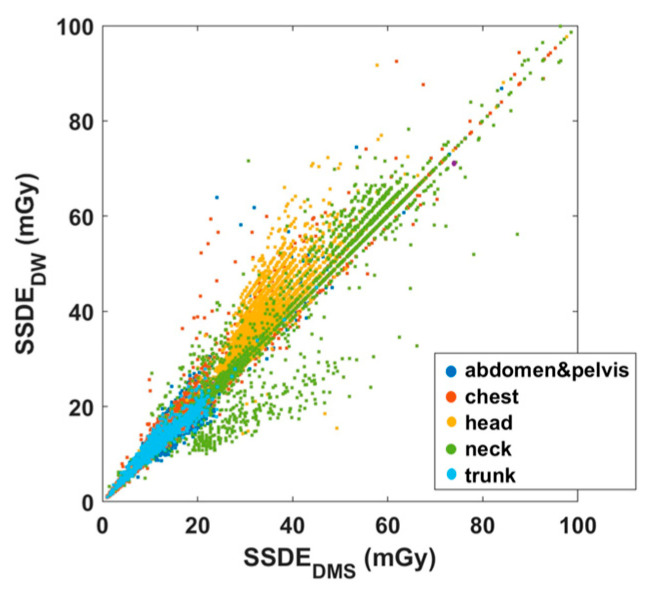
Scatter plots illustrating the correlations of SSDE_DW_ and SSDE_DMS_. The body regions are indicated with dots of different colors.

**Figure 5 diagnostics-15-01654-f005:**
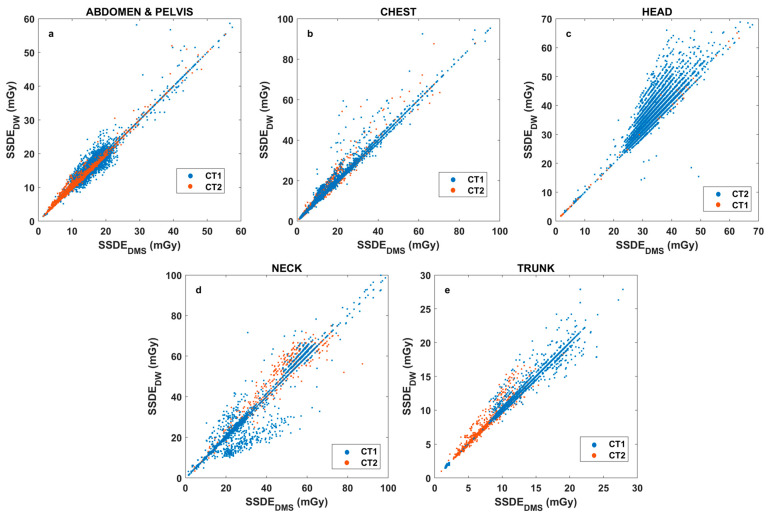
Scatter plots on region-based SSDE correlation data for both CT scanners (Key: blue dots—CT1 scanner, orange dots—CT2 scanner). (**a**): Abdomen – pelvis; (**b**): chest; (**c**): head; (**d**): neck; (**e**): trunk.

**Figure 6 diagnostics-15-01654-f006:**
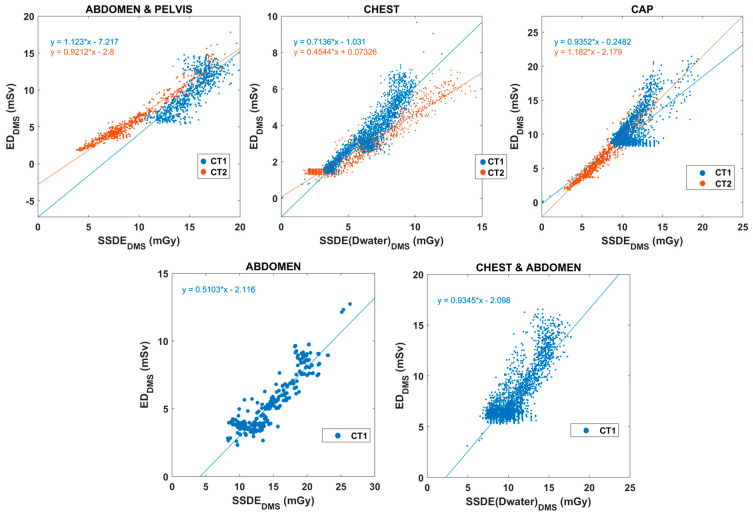
Relationships between SSDE_DMS_ and ED_DMS_ for different body regions. The two colors stand for CT1 and CT2 scanners (CAP: chest-abdomen-pelvis). There were no abdomen and chest-abdomen CT exams from the CT2 scanner.

**Figure 7 diagnostics-15-01654-f007:**
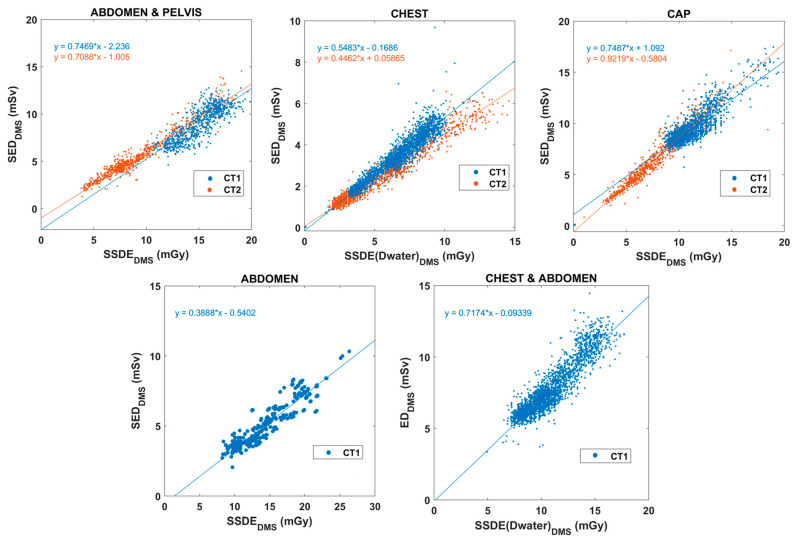
Correlation plots of SSDE_DMS_ and SED_DMS_ for both CT scanners and different body parts (CAP: chest-abdomen-pelvis). There were no abdomen and chest-abdomen CT exams from the CT2 scanner.

**Figure 8 diagnostics-15-01654-f008:**
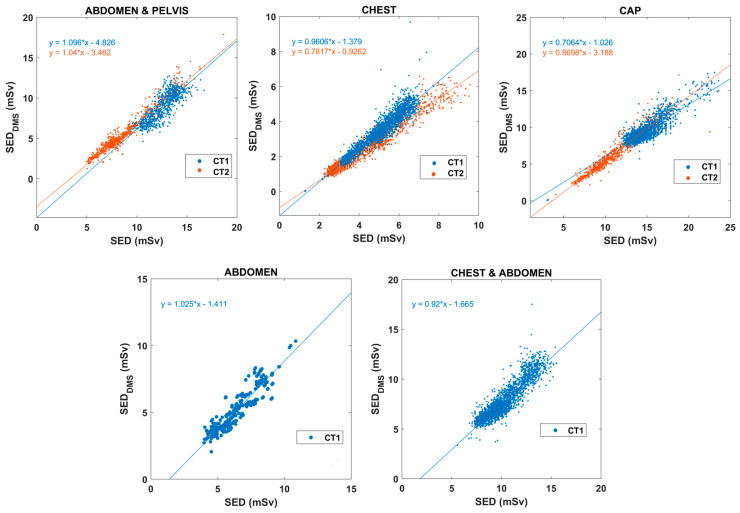
Correlations between SED_DMS_ and SED values across different body regions for CT1 and CT2 scanners (Key: SED—size-specific effective dose; DMS—dose-monitoring system; ED—effective dose).

**Figure 9 diagnostics-15-01654-f009:**
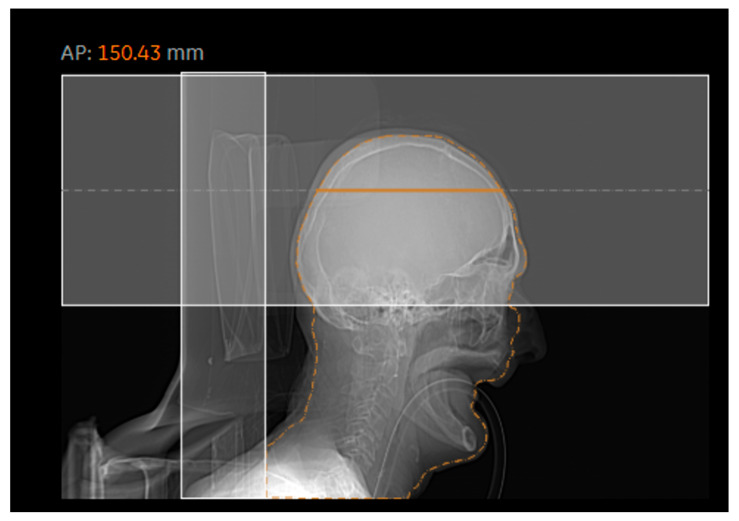
Representative image showing an inadequately defined central slice by DW^TM^.

**Table 1 diagnostics-15-01654-t001:** Region nomenclature based on the work of Martin and co-workers [11]. Detailed list of the scanned regions with corresponding scan length and the number of examined cases in our study.

Scanned Region	Scan Length (cm)	Number of CT Exams
Abdomen-pelvis	40–50	1779
Chest	30–37	2490
Chest-abdomen-pelvis (CAP)	62–70	2259
Abdomen	15–30	366
Chest-abdomen	40–50	2784

**Table 2 diagnostics-15-01654-t002:** Linear fitting results for ED_DMS_-SSDE_DMS_ and SED-SSDE data for different body regions and scanners. The number of CT exams, scan lengths, and the fitted parameters “a”, “b”, and the related R^2^—corresponding to the fitted equation of y = ax + b—are presented separately for each region. The table also includes the “a” and “b” parameters from Martin et al. [11]. The absolute percentage differences between the slopes for SED_DMS_-SSDE_DMS_ and ED_DMS_-SSDE_DMS_ correlations (compared to the SED-SSDE slopes) are stated in parentheses next to the “a” values.

		SED_DMS_ as Function of SSDE_DMS_		ED_DMS_ as Function of SSDE_DMS_		SED as Function of SSDE
Scanned Region		CT1	CT2	CT1	CT2	Martin et al. [11]
Abdomen and pelvis	a	0.747 (10%)	0.709 (4%)	1.123 (65%)	0.921 (35%)	0.6813
	b	−2.236	−1.005	−7.217	−2.8	2.3621
	R^2^	0.675	0.874	0.657	0.936	0.939
Chest	a	0.548 (4%)	0.446 (22%)	0.714 (25%)	0.454 (21%)	0.5708
	b	−0.169	0.059	−1.031	0.0733	1.2599
	R^2^	0.905	0.949	0.850	0.927	0.972
Chest-abdomen-pelvis	a	0.749 (29%)	0.922 (13%)	0.935 (12%)	1.182 (12%)	1.0599
	b	1.092	−0.580	−0.248	−2.179	2.9980
	R^2^	0.759	0.913	0.452	0.921	0.987
Abdomen	a	0.389 (3%)		0.510 (34%)		−0.3793
	b	−0.540		−2.116		1.7078
	R^2^	0.844		0.827		0.949
Chest-abdomen	a	0.717 (8%)		0.935 (20%)		0.7798
	b	−0.093		−2.098		0.8491
	R^2^	0.818		0.673		0.980

## Data Availability

The data presented in this study are available in the article and the corresponding Appendix A.

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
