# Peer review of "Relationship Between Effective Dose, Alternative Metrics, and SSDE: Experiences with Two CT Dose-Monitoring Systems"

_diagnostics, 2025, doi:10.3390/diagnostics15131654_

Round 1
Reviewer 1 Report
Comments and Suggestions for Authors
In this application study, the authors demonstrated the differences in effective dose estimates between their in-house system and a commercial system, and estimated size-specific effective dose (SED) based on DLP and then SSDE of CT scans.
It sounds like the authors were trying to demonstrate their in-house DMS system by computing ED and SED and comparing with existing commercial DW system. Perhaps the authors should shift the focus of the paper towards that direction to create a more purposeful and meaningful manuscript. The flaws of DW system in discussion sounds more interesting than repeating the existing methods from literature, why not focus on the improvement of DMS system over DW system?
For DW, how did the authors assign region names to CT studies? Had the authors verified assignment or computed the accuracy of the assignment?
For DMS, how did the authors compute the ED for an entire study and is it different from the method by DW? For example, whether ED from scout scan is included.
Why is the number of involved series different in DW (79383) from in DMS (105941)?
How was the effective diameter determined in DMS system? Why not use water-equivalent diameter for all body regions in DMS system?
For CT1 and AP body region in table 2, why is the empirical equation for ED/SSDE so different from its SED/SSDE counterpart? What is the coefficient of determination (R2)? In addition, the equation is quite different from one scanner to another. Do the authors generate separate equations for each CT machine? If patient population shifted/changed, do the authors have to generate the equations again?
Page 2, line 66: Huda et al [12] did not mention weight-adjusted effective dose in this paper, please change the citation here.
Page 2, line 84: suggest changing “cancer risk” to “SED” because no size-specific conversion factor is available to convert SED to cancer risk.
Page 2, line 90: please be consistent with terminology for SED and try to make the nomenclature clear and concise.
Page 3, line 91: suggest changing “by what means and how accurately the SED parameter can be estimated from the SSDE values derived from real CT scans” to “the various methods and accuracy of SED estimation from the SSDE values derived from real CT scans”.
Page 4, line 97: suggest changing “(hereafter GE Revolution HD and GE Revolution Evo CT scanners are referred to as CT1 and CT2 respectively)” to “(referred to as CT1 and CT2 respectively)”.
Page 4, equation 4: suggest changing ED to SED in this equation and in text to distinguish it from ED.
Page 4, line 127: Deak et al [8] did not provide weight-specific conversion factors, but conversion factors based on representative phantoms of a few age groups. What factors did the authors used, and where did the authors apply the factors to?
Page 4, line 136: suggest changing “the CTDI and the suggested formula, which has been devised by the AAPM” to “the CTDIvol and the formula suggested by the AAPM”.
Suggest adding superscripted “TM” to all the commercial software mentioned in the text because these are registered trademarks.
Page 5, line 164: cite the equation numbers from Martin et al [16, 18] in the text, and show the equation in this paper.
Figure 1: Define what box and whiskers represents, e.g. box for 25th and 75th percentiles while whiskers for min and max.
Figure 2: suggest using different symbols for different body regions and increasing the size of the legend.
Page 7, line 195-200: what was the issue with DW system? Please clarify. How exactly does DW system compute ED for a study including series of different body regions, e.g. chest series and chest-abdomen-pelvis series?
Page 7, line 211-212: scan length is mostly irrelevant to SSDE and is only important to DLP. CTDIvol for head is much higher than CTDIvol for body, and thus SSDE is usually higher for head than for body. Please improve the understanding of the concepts used in this study, including SSDE, CTDIvol and DLP.
Figure 4: increase the size of the legend.
Figure 5: again, SSDE is irrelevant to scan length and showing it in figure is not providing any useful information. For the right figure, DLP = scan length x CTDIvol and the positive correlation is expected.
Table 2: cite Martin’s name in the table instead of “reference 16”.
Page 11, line 280: what is “less consistent with the EDDMS_kg data”?
Page 11, line 295-296: why not exclude the protocols including different body regions? There were thousands of patient studies involved in this paper, right?
Page 11, line 314-320: in this paragraph, the authors pointed out SSDE was unaffected by scan length in the beginning and then concluded SSDE depended on scan length in the end. So, does the SSDE depend on scan length or not?
Page 12, line 323: why would SSDE be a surrogate for SED? Yes, they are linearly correlated with perhaps nice R2 values, but they are not 1:1 and the ratio changes between body regions and scanners. Knowing SSDE of 10 mGy does not guarantee a SED of 10 mSv.
Page 12, line 342: where does the number 70,000 CT images come from? Please be consistent within the manuscript.
Page 12, line 356-358: where did the authors show weight-corrected ED is better than ED? In figure 1 there was little visual difference, and the authors did not provide a statistical test result showing if there is a significant difference, letting alone showing one is superior to the other.
Reviewer 2 Report
Comments and Suggestions for Authors
The authors present a comparison of different values for estimating the effective dose caused by CT applications.
In a first step, they check the accuracy of a local developed software tool by a commercial tool. However, this comparison is limited because the tools use different conversion factors. Therefore, in special cases the results differ between the DW and the DMS tool by a factor 5 (see fig. 2c). These deviations should be discussed in more detail (Are they only caused by the different conversation factors?). In a second step, the authors try to study the differences of the introduced dose values for two different CT scanner, but the differences of these scanners are obviously too large in patient collective and body regions. In the third step, the authors investigate the correlations of the different dose values to the patient size-specific effective dose and their prognostic potential for the real patient effective dose. The authors should discuss, why they prefer SSDE although EDDMS_kg better correlates to SED. The reason could be that SSDE is easier to evaluate.
Some further remarks:
- line 126: It is a repetition of line 116.
- line 167: chest-abdomen is double.
- line 175: SED instead of SDE.
- lines 311-312: DRL instead of DLR.
- text of fig 6.: are the data from studies or series?
- line 222-226: SSDE and ED should have the index DMS.
- line 228: fig. 5b is “better” than 5a, but it is only a very weak “positive relationship”.
- lines 136-141: the capital “D” stands usually for dose. Maybe, “d” is better for diameter.
- The DMS conversion factors come in line 117 from AAMP, in line 126 and the equations from RPT 96 and in line 286 from ICRP but is always the same source. A unique formulation would be less confusing.
- The dots in the legends of the figures are very small. Therefore, it is difficult to recognize the colour.
Author Response
Dear Reviewer 2,
Thank you for the review and remarks in relation to our work. Please find the answers to the questions and comments below. The text was throughougly reviewed, significant grammatical and stylistic corrections were performed. All modifications and supplementations made in the main text can be found in the "track changes" version of the manuscript.
The authors present a comparison of different values for estimating the effective dose caused by CT applications.
In a first step, they check the accuracy of a local developed software tool by a commercial tool. However, this comparison is limited because the tools use different conversion factors. Therefore, in special cases the results differ between the DW and the DMS tool by a factor 5 (see fig. 2c). These deviations should be discussed in more detail (Are they only caused by the different conversation factors?). In a second step, the authors try to study the differences of the introduced dose values for two different CT scanner, but the differences of these scanners are obviously too large in patient collective and body regions. In the third step, the authors investigate the correlations of the different dose values to the patient size-specific effective dose and their prognostic potential for the real patient effective dose. The authors should discuss, why they prefer SSDE although EDDMS_kg better correlates to SED. The reason could be that SSDE is easier to evaluate.
"Are they only caused by the different conversation factors?"
Response: In order to find out how the different factors influence the ED, in the case of the abdomen, chest and pelvis, we determined the slope of the associated populations from Fig 2c., and then the ratio of their corresponding f factors:
|
Regions |
Slope |
fdw/fdms |
Difference |
|
Abdomen |
0.985 |
0.98 |
0.005 |
|
Pelvis |
0.857 |
0.86 |
0.003 |
|
Chest |
0.971 |
0.959 |
0.012 |
Overall, we conclude that slopes are well-matched with the ratio of f-factors.
Some further remarks:
line 126: It is a repetition of line 116.
line 167: chest-abdomen is double.
line 175: SED instead of SDE.
lines 311-312: DRL instead of DLR.
Response: We carefully looked through the text and corrected the typos and the duplications.
text of fig 6.: are the data from studies or series?
Response: As for Fig. 6, we applied data from DMS, and the ED values were available for each series.
line 222-226: SSDE and ED should have the index DMS.
Response: Thank you for the comment. We removed Fig. 5 and the related sentences (lines 222–226) from the manuscript because the demonstration of the uncorrelated SSDE-scan length relationship provides no additional information.
line 228: fig. 5b is "better" than 5a, but it is only a very weak "positive relationship".
Response: Thank you for the comment. We removed Fig. 5 from the manuscript. Please find the response above.
lines 136-141: the capital "D" stands usually for dose. Maybe, "d" is better for diameter.
Response: Thank you for the comment. Based on prior works this seems the frequently used abbreviation (e.g., Ref. 16), that is reason why we would rather choose this form.
The DMS conversion factors come in line 117 from AAMP, in line 126 and the equations from RPT 96 and in line 286 from ICRP but is always the same source. A unique formulation would be less confusing.
Response: The reference to the ICRP was incorrect; it should have been to the AAPM. We have corrected this in the revised manuscript.
The dots in the legends of the figures are very small. Therefore, it is difficult to recognize the colour.
Response: Based on the Reviewer's suggestion we increased the dots in the legends.
Reviewer 3 Report
Comments and Suggestions for Authors
The manuscript presents a comprehensive comparison of two CT dose monitoring systems, offering valuable insights into their effectiveness and limitations. The work is well-structured, methodologically sound, and supported by a large dataset. However, to improve clarity and scientific robustness, I suggest the following minor revisions:
1) The total number of CT scans analyzed is mentioned in the Methods section, but the exact case numbers should also be explicitly included in the captions of Figures and Supplementary Tables (e.g., Table S1–S3) to improve transparency.
2) It is unclear whether the SSDE values shown in Figure 1 are derived from DMS or DW. A brief clarification in the figure caption would help avoid misinterpretation.
3) In Supplementary Figure S1b, no apparent association is observed between SSDE and scan length. This should be briefly explained in the main text, particularly emphasizing the role of bolus tracking scans in this observation.
4) In the discussion of Figure 2, the manuscript mentions that DW assigns the same region to all series in a study. It would be helpful to clarify that this assignment is fixed and non-editable, which can lead to overestimation or underestimation of ED in multi-region scans.
5) The fitted parameters presented in Table 2 are compared with those of Martin et al., but the differences are not explicitly discussed. Consider adding numerical comparisons (e.g., percentage deviation) to support your interpretation.
6) Supplementary Figures S2 through S5 illustrate critical issues in region labeling and ED estimation. These should be explicitly referenced in the Results or Discussion sections to guide readers to the supporting visuals.
Author Response
Dear Reviewer 3,
Thank you for the review and remarks in relation to our work. Please find the answers to the questions and comments below. The text was throughougly reviewed, significant grammatical and stylistic corrections were performed. All modifications and supplementations made in the main text can be found in the "track changes" version of the manuscript.
The manuscript presents a comprehensive comparison of two CT dose monitoring systems, offering valuable insights into their effectiveness and limitations. The work is well-structured, methodologically sound, and supported by a large dataset. However, to improve clarity and scientific robustness, I suggest the following minor revisions:
- The total number of CT scans analyzed is mentioned in the Methods section, but the exact case numbers should also be explicitly included in the captions of Figures and Supplementary Tables (e.g., Table S1–S3) to improve transparency.
Response: Thank you for the comment. We revised the supplementary material according to the suggestions of the Reviewer.
- It is unclear whether the SSDE values shown in Figure 1 are derived from DMS or DW. A brief clarification in the figure caption would help avoid misinterpretation.
Response: Thank you for the comment. The figure caption was supplemented with an explanatory text (please see the updated Figure in the revised version of the manuscript)
- In Supplementary Figure S1b, no apparent association is observed between SSDE and scan length. This should be briefly explained in the main text, particularly emphasizing the role of bolus tracking scans in this observation.
Response: We absolutely agree with the Reviewer; no correlation is expected between SSDE and scan length. We only intended to demonstrate this fact, but since both You and the other Reviewer found the discussion of this relationship unnecessary, we removed these parts from the manuscript and the supplementary doc.
- In the discussion of Figure 2, the manuscript mentions that DW assigns the same region to all series in a study. It would be helpful to clarify that this assignment is fixed and non-editable, which can lead to overestimation or underestimation of ED in multi-region scans.
Response: Based on the Reviewer's comment, we included Figure S2 (now please see as Fig. 3) into the manuscript to illustrate the limitations of DoseWatch (specifically, those of the currently used version: 3.1.5) and highlighted the definition of the non-editable region.
- The fitted parameters presented in Table 2 are compared with those of Martin et al., but the differences are not explicitly discussed. Consider adding numerical comparisons (e.g., percentage deviation) to support your interpretation.
Response: We appreciate the comment. Accordingly, we supplemented Table 2 with percentage differences and R² values. In addition, related conclusions were added to the Discussion section as well.
- Supplementary Figures S2 through S5 illustrate critical issues in region labeling and ED estimation. These should be explicitly referenced in the Results or Discussion sections to guide readers to the supporting visuals.
Response: Thank you for the suggestion. We moved one of these figures (the original Figure S1) into the Results section and also referenced the updated Figures S1-S3 in the manuscript.
Round 2
Reviewer 1 Report
Comments and Suggestions for Authors
Greatly improved manuscript! But there are still a few things to be addressed:
Abstract, conclusion: suggest changing “its additive property” to “its size-specific property”.
Table 2: Instead of using “vs”, suggest showing SED and ED as a function of SSDE in the top row for better clarity, e.g. “SED (SSDE)” or “SED as a function of SSDE” and append subscript as appropriate.
Introduction, final sentence: for better clarity, suggest changing to: Finally, we benchmarked the estimated SEDDMS against accurate SED generated by Martin and colleague with Monte Carlo simulation[reference].
Line 480: suggest adding references for the f factors used in DMS and DW if possible.
Discussion, first paragraph: when mentioning Monte Carlo simulation here, please cite Martin’s paper as the authors did not perform such simulations in this study.
Line 960-964: consider showing average (min-max) of absolute percentage differences for ED related parameters and for SED related parameters to make the point that SED can be more accurately estimated than ED. Or, since “b” still affects “y” albeit at a smaller scale, maybe the authors should consider using a sample (e.g. 20 studies) of SSDE for each scanned region to compute ED or SED, and then compute the average (min-max) difference between proposed method and Martin et al. method, instead of showing differences of one parameter of the fitting equation.
Comments on the Quality of English LanguagePlease consider the suggested language edits. If possible, the authors can ask an English-speaking colleague or friend to proofread the manuscript.
Author Response
Thank you for the review and remarks in relation to our work. Please find the answers to the questions and comments below. The text was throughougly reviewed, significant grammatical and stylistic corrections were performed. All modifications and supplementations made in the main text can be found in the "track changes" version of the manuscript.
Greatly improved manuscript! But there are still a few things to be addressed:
1. Abstract, conclusion: suggest changing “its additive property” to “its size-specific property”.
Response: In accordance with the reviewer’s suggestion, we requested an American professor to proofread the manuscript in order to enhance the quality of the English. As a result, the following sentences have been revised accordingly:
„ SED is a suitable supplementary size-specific dose data to SDDE and may be a preferable choice for estimating cumulative doses in routine radiological practice.”
2. Table 2: Instead of using “vs”, suggest showing SED and ED as a function of SSDE in the top row for better clarity, e.g. “SED (SSDE)” or “SED as a function of SSDE” and append subscript as appropriate.
Response: We agree with the Reviewer. We made the modifications according to the Reviewer's recommendation. We changed “vs” to “as function of” in the table.
3. Introduction, final sentence: for better clarity, suggest changing to: Finally, we benchmarked the estimated SEDDMS against accurate SED generated by Martin and colleague with Monte Carlo simulation[reference].
Response: In accordance with the reviewer’s suggestion, we requested an American professor to proofread the manuscript in order to enhance the quality of the English. As a result, the following sentences have been revised accordingly:
„Third, compare SEDDMS against SED generated from the Monte Carlo simulation reported by Martin et al. [16].”
4. Line 480: suggest adding references for the f factors used in DMS and DW if possible.
Response: The quoted line has been modified: “The small difference between the total doses (19.24 vs 19.96.) is explained by the distinct f factors used in DWTM [8] and DMS [15].”
References:
[8]: Deak, P.D.; Smal, Y.; Kalender, W.A. Multisection CT protocols: sex- and age-specific conversion factors used to determine effective dose from dose-length product. Radiology 2010, 257 (1), 158-66. https://doi.org/10.1148/radiol.10100047
[15]: AAPM Report 96: [ Medicine, A. A.o. P.i. AAPM Report No. 96. The measurement, reporting, and management of radiation dose in CT. (College Park, MD: American Association of Physicists in Medicine) 2008
5. Discussion, first paragraph: when mentioning Monte Carlo simulation here, please cite Martin’s paper as the authors did not perform such simulations in this study.
Response: In accordance with the reviewer’s suggestion, we requested an American professor to proofread the manuscript in order to enhance the quality of the English. As a result, the following sentences have been revised accordingly.
6. Line 960-964: consider showing average (min-max) of absolute percentage differences for ED related parameters and for SED related parameters to make the point that SED can be more accurately estimated than ED. Or, since “b” still affects “y” albeit at a smaller scale, maybe the authors should consider using a sample (e.g. 20 studies) of SSDE for each scanned region to compute ED or SED, and then compute the average (min-max) difference between proposed method and Martin et al. method, instead of showing differences of one parameter of the fitting equation.
Response: We have supplemented the part of the Discussion referring to Table 2 with the following sentence: “We observed variable correlation (0.452-0.949) and percentage differences between ED and SSDE (Table 2) suggesting the discrepancies are not fundamentally related to the variance of data points. The mean (and the range) absolute percentage difference between EDDMS-SSDEDMS was always higher at about 28% (12%-65%) than between SEDDMS–SSDEDMS [11.6% (4%-29%)].”
Additionally, by now, the Table 2. presents the absolute values of the percentages.
We appriciate the Reviewer’s work and comments. Trusting in your positive evaluation.
Yours sincerely,
Lilla Szatmáriné Egeresi